# Transcriptomic Profiling of Shoot Apical Meristem Aberrations in the Multi-Main-Stem Mutant (*ms*) of *Brassica napus* L.

**DOI:** 10.3390/genes14071396

**Published:** 2023-07-03

**Authors:** Qian Wang, Na Xue, Chao Sun, Jing Tao, Chao Mi, Yi Yuan, Xiangwei Pan, Min Gui, Ronghua Long, Renzhan Ding, Shikai Li, Liangbin Lin

**Affiliations:** 1College of Agronomy and Biotechnology, Yunnan Agricultural University, Kunming 650201, China; wq@yaas.org.cn (Q.W.); 2020110024@stu.ynau.edu.cn (C.M.); pxw0916@163.com (X.P.); 2Horticultural Research Institute, Yunnan Academy of Agricultural Sciences, Kunming 650200, China; xuena0129@163.com (N.X.); taojing1101@163.com (J.T.); lareine.1213@163.com (Y.Y.); mingui6799@126.com (M.G.); longronghua134@sohu.com (R.L.); dingrenzhan@139.com (R.D.); 3Engineering Research Center of Vegetable Germplasm Innovation and Support Production Technology, Horticultural Research Institute, Yunnan Academy of Agricultural Sciences, 2238 Beijing Road, Kunming 650205, China; 4Tea Research Institute, Yunnan Academy of Agricultural Sciences, Kunming 650221, China; sunchaoynau@163.com

**Keywords:** *Brassica napus* L., shoot apical meristem (SAM), shoot branching, transcriptome, cytokinin

## Abstract

Rapeseed (*Brassica napus* L.) is a globally important oilseed crop with various uses, including the consumption of its succulent stems as a seasonal vegetable, but its uniaxial branching habit limits the stem yield. Therefore, developing a multi-stem rapeseed variety has become increasingly crucial. In this study, a natural mutant of the wild type (ZY511, Zhongyou511) with stable inheritance of the multi-stem trait (*ms*) was obtained, and it showed abnormal shoot apical meristem (SAM) development and an increased main stem number compared to the WT. Histological and scanning electron microscopy analyses revealed multiple SAMs in the *ms* mutant, whereas only a single SAM was found in the WT. Transcriptome analyses showed significant alterations in the expression of genes involved in cytokinin (CK) biosynthesis and metabolism pathways in the *ms* mutant. These findings provide insight into the mechanism of multi-main-stem formation in *Brassica napus* L. and lay a theoretical foundation for breeding multi-main-stem rapeseed vegetable varieties.

## 1. Introduction

Rapeseed is one of the world’s most important oil crops, accounting for approximately 15% of the world’s vegetable oil production [1]. As the social economy develops and consumers demand more diversified products, rapeseed is becoming increasingly versatile, serving as a vegetable, a flower, a feed for bees that produce honey, fodder, and a fertilizer [1,2]. Although the crispy and succulent bolting stem of rapeseed is a seasonal vegetable commonly consumed in South China, the stem yield is lower than that of other crops, such as Chinese flowering cabbage (*Brassica campestris* L. ssp. *Chinensis* var. *utilis Tsen et Lee*) and Chinese kale (*Brassica alboglabra*) due to its single main stem. Therefore, exploring or creating new multi-main-stem rapeseed varieties could effectively enhance vegetable yield.

Branch and stem formation in plants is facilitated by the differentiation of axillary meristems (AMs) at leaf axils, which elongate and differentiate into lateral branches [3,4]. The AM initiates from the shoot apical meristem (SAM), a group of pluripotent stem cells that generate leaf, axillary branch, flower primordium, and stem tissue [5,6]. Thus, the final number of branches or stems is determined by the number of SAMs. Phytohormones, including auxin, cytokinins, and strigolactones, play an important role in activating, initiating, and regulating plant branching [7,8]. Among these, cytokinins (CKs) have been demonstrated to be positive regulators of branching and SAM development [9,10]. CKs are small molecules derived from adenine and modified with an isoprenoid or aromatic side chain at the N^6^ position [11]. The active forms of CKs, which are most widespread, include four isoprenoid types known as natural isoprenoid CKs or CK nucleobases: isopentenyladenine (iP), trans-zeatin (tZ), cis-zeatin (cZ), and dihydrozeatin (DHZ) [12,13]. These CK nucleobases can undergo various conversions to form different CK conjugates. Notably, they can be transformed into CK ribosides (tZR, cZR, DZR, and iPR), CK phosphates (tZRPs, cZRPs, DZRPs, and iPRPs), CK O-glucosides (tZOG, cZOG, tZROG, and cZROG), and CK N-glucosides (tZ7G, tZ9G, iP7G, and iP9G) [14,15]. Moreover, aromatic CKs, such as benzyladenine (BA), ortho-topolin (oT), meta-topolin (mT), and their methoxy-derivatives (meoT and memT), have gained significant attention in bioassays due to their remarkable biological activity [16,17].

Nishimura et al. (2004) used the Arabidopsis *ahk2 ahk3 ahk4* triple CK receptor mutant and revealed that CKs function in the maintenance of SAM activity [18]. In Arabidopsis, the *supershoot* (*sps*) mutant forms multiple AMs in the leaf axils due to increased CK levels, promoting branch formation from rosette and cauline leaves [19]. Moreover, the loss of the negative feedback regulator of CK, type-A ARR homolog *ABPH1*, results in an increase in SAM size in maize [20]. Undoubtedly, cytokinin (CK) signaling plays a pivotal role in the regulation of WUSCHEL (WUS), a key regulator controlling SAM activity [21]. The induction of *WUS* transcription by CK leads to its upregulation within the SAM [22]. In Arabidopsis, the type-B Arabidopsis Response Regulators (ARRs), including ARR1, play a direct role in binding to the promoter region of *WUS*, thereby facilitating its transcriptional activation [23,24]. Furthermore, type-B ARRs have the ability to activate the expression of type-A ARRs, such as *ARR7* and *ARR15*, which act as negative regulators of CK signaling [25]. Notably, WUS forms an additional positive feedback loop with CK signaling by directly suppressing the transcription of several type-A ARRs [26]. This intricate interplay between CK and WUS ensures a precise balance between the maintenance of stem cells and their differentiation during plant development.

The biosynthesis and catabolism of CK are dynamically balanced processes in plants. Cytokinin oxidase (CKX) can irreversibly cleave active CKs, decreasing cytokinin levels [27]. CKX family genes have been identified in many plant species, such as *Arabidopsis thaliana* [28], *Nicotiana tabacum* [29], *Zea mays* [30], *Oryza sativa* [31], *Triticum aestivum* [32], and *Glycine max* [33]. In rapeseed, 23 *CKX* genes were identified [34]. *CKX* has been found to play a role in SAM development and plant branching. Werner et al. (2003) found that the Arabidopsis *CKX* gene family has seven members (At*CKX1* to At*CKX7*), and transgenic plants overexpressing *CKX1* and *CKX3* showed decreased SAM activity [28]. *LONELY GUY* (*LOG*) encodes a cytokinin riboside 5′-monophosphate phosphoribohydrolase that functions in the final step of bioactive CK synthesis; the analysis of a rice (*Oryza sativa*) mutant in which *lonely guy* (*log*) was mutated showed that it is important for the termination of shoot meristems [35]. In Arabidopsis, nine *LOG* genes (At*LOG1* to At*LOG9*) were predicted to be homologs of rice *LOG* [36]. Tokunaga et al. (2011) used a multiple *Atlog* mutant and found that the *LOG* genes were important for the maintenance of the SAM in Arabidopsis [37].

Multi-main-stem (MMS) traits have been discovered in rice and wheat, which improve the growth potential and seed number and are therefore important for yield improvement [38,39]. Zhao et al. (2019) identified six candidate genes (*RPT2A*, *HLR*, *CRK*, *LRR-RLK*, *AGL79*, and *TCTP*) involved in SAM differentiation and axillary bud formation by QTL mapping and the gene-fishing technique, which were related to the formation of the MMS phenotype in rapeseed [40]. However, the mechanism underlying MMS formation and the relationship between SAM and MMS formation remain unclear. In this study, we identified a rapeseed mutant with MMS from ZY511 (WT, Zhongyou511), and after six consecutive generations with self-selection, we generated lines with stable inheritance of the multiple main branching trait. The mutant, named Multiple Stem (*ms*) rapeseed, develops over six main stems from the base of the plant, with small branching angles. Our results showed that the development of SAM in the *ms* mutant was abnormal compared to that in the WT. Furthermore, through transcriptome analysis conducted at various stages of germination, we discovered that the aberrant development of the SAM in the *ms* mutant initiates at 20 days after germination (DAG). Notably, within the SAM of the *ms* mutant, we observed significant alterations in the expression of genes associated with the CK signaling pathway. These findings establish a solid foundation for unraveling the molecular mechanisms underlying the formation of multiple main stems and provide reference for the selection and breeding of rapeseed varieties with multiple main stems.

## 2. Methods and Materials

### 2.1. Plant Materials and Growth Conditions

Wild-type and *ms* mutant seeds were sterilized in 75% ethanol for 15 min and then rinsed repeatedly in sterile water. The seeds were germinated on Petri dishes containing Murashige Skoog (MS) medium (see Appendix A for the recipe) in an incubator at 25 °C and were sown in culture bottles once they germinated. The bottles were placed in a light incubator with 14 h of light, 10 h of dark, a day temperature of 22 °C, a night temperature of 18 °C, and a relative humidity of 60%, with a light intensity of 10,000 lux.

### 2.2. Tissue Section

To prepare paraffin sections, the SAM tissues of wild-type and *ms* mutant plants at 20 days after germination (DAG) were cut and then placed in a stationary formaldehyde acetic acid alcohol (FAA) solution for 24 h, which was performed by Wuhan Servicebio Technology Co., Ltd. (Wuhan, China). Please refer to [41,42] for further details. The specific steps are described below.

#### 2.2.1. Paraffin Section Preparation

The tissue was removed from the fixing solution, the target tissue was trimmed using a scalpel in a ventilated cupboard, and the trimmed tissue was labeled and put in a dehydration box.

The dehydration box was placed in a dehydrator (Diapath Donatello, Milan, Italy) for dehydration with an alcohol gradient: 75% alcohol for 4 h, 85% alcohol for 2 h, 90% alcohol for 2 h, 95% alcohol for 1 h, anhydrous ethanol for 1 h, alcohol benzene for 5~10 min, xylene for 10~20 min, and 65 °C molten paraffin for 3 h.

The wax-soaked tissue was embedded in an embedding machine (WHJJ JB-P5, Wuhan, China). First, the melted wax was placed in the embedding frame, and before the wax solidified, the tissue was removed from the dewatering box and put into the embedding frame according to the requirements of the embedding surface, and the corresponding label was affixed. This was then cooled at −20 °C on a freezing table (WHJJ JB-L5, Wuhan, China), and after the wax had solidified, the wax block was removed from the embedding frame and repaired by using a scalpel. This process ensured the production of clean and precise sections, which were suitable for subsequent experiments.

The trimmed wax block was placed on a −20 °C freezing table, and the modified tissue chip wax block was sliced at a thickness 4 μm using a paraffin slicer (Leica RM2016, Wetzlar, Germany). The slice of tissue was floated on 40 °C warm water in a spreading machine to flatten it (KEDEE KD-P, Jinhua, China), and the tissue was then picked up using glass slides and baked in an oven at 60 °C. After the water-baked dried wax had melted, it was taken out and stored at room temperature.

#### 2.2.2. Paraffin Section Safranin O-Fast Green Staining

The sections were rehydrated in BioDewax and Clear Solution (Servicebio, Wuhan, China) for 40 min, 100% ethanol for 10 min, and 75% ethanol for 5 min. Finally, they were rinsed under running water.

The sections were placed in safranin O staining solution for 2 h, and then into 50%, 70%, and 80% ethanol, each for 3~8 s. The sections were then put into Fast Green staining solution (Servicebio, Wuhan, China) for 6~20 s, and dehydrated by washing three times with 100% ethanol for 5 min each. Finally, the tissue sections were mounted using neutral balsam. They were observed and photographed using an ECHO microscope (Revolve FL, San Diego, CA, USA).

### 2.3. Scanning Electron Microscopy

The SAM samples of 20 DAG wild-type and *ms* mutant plants were used for scanning electron microscopy (SEM) analysis, which was performed by Wuhan Servicebio Technology CO., Ltd. (Wuhan, China), using a Hitachi SU8100 SEM. The procedures are detailed below.

The target fresh tissues were selected in a manner that minimized mechanical damage such as pulling, contusion, and extrusion. A sharp blade was used to quickly cut and harvest the fresh tissue blocks, within 1–3 min. The area of the tissue block was to be no more than 3 mm^2^. The tissues were then gently washed with PBS. The target side of the tissue was labeled (the side that was to be observed). Care was taken to protect the tissue blocks, especially the target side, from mechanical damage such as extrusion with the forceps. The washed tissue blocks were immediately fixed using an electron microscopy fixative (Servicebio, Wuhan, China) for 2 h at room temperature, and then transferred into 4 °C storage for preservation and transportation.

Tissue blocks were washed with 0.1 M PB (pH 7.4) 3 times, for 15 min each time. Then, the tissue blocks were transferred into 1% OsO4 in 0.1 M PB (pH 7.4) for incubation for 1–2 h at room temperature. After that, the tissue blocks were washed in 0.1 M PB (pH 7.4) 3 times, for 15 min each time.

The samples were dehydrated as follows: 30% ethanol for 15 min; 50% ethanol for 15 min; 70% ethanol for 15 min; 80% ethanol for 15 min; 90% ethanol for 15 min; 95% ethanol for 15 min; two changes of 100% ethanol for 15 min; and isoamyl acetate for 15 min.

The samples were dried using a critical point dryer (Quorum K850, Rockville, ML, USA). The specimens were attached to metallic stubs using carbon stickers and sputter-coated (Hitachi MC1000, Tokyo, Japan) with gold for 30 s. They were observed and imaged using a scanning electron microscope (Hitachi SU8100, Tokyo, Japan).

### 2.4. Quantification of Cytokinins

The cytokinins in the SAMs of the wild-type and *ms* mutant plants at 20 DAG were quantified using HPLC-MS/MS (liquid chromatography–mass spectrometry) by Shanghai Applied Protein Technology Co., Ltd. (Shanghai, China), following a previously described method [43]. The SAMs from five individual replicate plants were pooled as one replicate, *n* = 3. The specific steps were as follows.

#### 2.4.1. Sample Preparation

The samples were taken out at −80 °C. After grinding with liquid nitrogen, 100 mg samples were weighed, and 1170 µL of an acetonitrile (CAN)/water/formic acid (FA) solution was added (80:19:1, *v*/*v*). Then, 20 µL of Internal Standard (IS) was added, and the mixture was vortexed for 60 s, exposed to ultrasound at low temperature in the dark for 25 min, left to stand at −20 °C overnight, and centrifuged at 14,000 rcf at 4 °C for 20 min. Subsequently, 900 µL of the supernatant was placed in a 25 mg 96-well descaling plate for positive pressure filtration. Then, 200 µL of an ACN/water/FA solution (80:19:1, *v*/*v*) was added for elution. The supernatant was dried in liquid nitrogen. For LC-MS analysis, the samples were redissolved in 200 μL of MeOH/water (1:1, *v*/*v*), adequately vortexed, and then centrifuged (14,000 rcf, 4 °C, 15 min). The supernatants were collected for LC-MS/MS analysis.

#### 2.4.2. HPLC-MS/MS Analysis

HPLC analysis: Analyses were performed using an UHPLC (1290 Infinity LC, Agilent Technologies, Santa Clara, CA, USA) coupled to a QTRAP (AB Sciex 5500, Sciex, Framingham, MA, USA). The mobile phase contained A: 0.05% FA in water and B: 0.05% FA in ACN. The samples were placed in the automatic sampler at 4 °C, and the column temperatures were kept constant at 45 °C. The gradients were at a flow rate of 400 µL/min, and a 4 µL aliquot of each sample was injected. Gradient B changed from 2% to 10% over 0–1 min, increased to 70% over 1–10 min, and then increased to 95% in 1 min; then, the proportion of B was reduced to 2% for 0.1 min and kept constant for 11–13 min. The QC samples used for testing and evaluating the stability and repeatability of this system, at the same time, set the standard mixture of metabolites, were also used for the correction of the chromatographic retention time.

MS/MS analysis (MRM): In ESI positive mode, the conditions were set as follows: source temperature: 550 °C; Ion Source Gas1 (Gas1): 55; Ion Source Gas2 (Gas2): 50; Curtain gas (CUR): 30; ionSapary Voltage Floating (ISVF): +4500 V. In ESI negative mode, the conditions were set as follows: source temperature: 550 °C; Ion Source Gas1 (Gas1): 55; Ion Source Gas2 (Gas2): 50; Curtain Gas (CUR): 30; ionSapary Voltage Floating (ISVF): −4500 V. The MRM mode detection ion pair was adopted.

### 2.5. RNA Extraction, Sequencing, and Analysis

The SAMs of wild-type and *ms* mutant plants at 15, 20, 25, 30, and 35 DAG were collected, and samples from five individual replicate plants were pooled as one replicate, *n* = 3. Samples were stored at −80 °C until use. Total RNA extraction and transcriptome sequencing were conducted as previously described. Briefly, total RNA was extracted using TRIZOL^®^ Reagent (TRAN, Beijing, China) according to the manufacturer’s protocol and quality-checked using the QubitTM4 Fluorometer microvolume spectrophotometer (Thermo Fisher Scientific, Singapore). cDNA libraries were then prepared and sequenced on the Illumina HiSeq4000 sequencing platform by Lian Chuan Biotechnology Co., Ltd. (Hangzhou, China). Quality control for the raw RNA-seq data from the machine was performed using the fastQC v0.11.2 software. Low-quality reads and adapter sequences were deleted using Trimmomatic (0.36.5) to acquire clean, high-quality reads [44]. The obtained clean reads were mapped to the published reference genome (http://www.genoscope.cns.fr/brassicanapus/, 15 January 2022). StringTie (1.3.4) was employed to count the number of reads mapped onto each gene, and gene expression was quantified as the number of fragments per kilobase of the transcript sequence per million base pairs (FPKM). Differential expression analysis was performed using the DESeq2R package (2.11.38).

Transcripts with *p* values < 0.05 and |log_2_(fold change)| ≥ 1 were considered differentially expressed genes (DEGs). Pearson correlation analysis, principal component analysis (PCA), and Venn diagram analysis and graphing were performed using the OmicShare tools (https://www.omicshare.com/tools, accessed on 18 February 2022). GO enrichment analysis was performed using AgriGO (http://systemsbiology.cau.edu.cn/agriGOv2/index.php, accessed on 19 February 2022). KEGG analysis was conducted with reference to the Kyoto Encyclopedia of Genes and Genomes (https://www.genome.jp/kegg/, accessed on 20 February 2022) database, and KOBAS was used to detect the statistically significant enrichment of DEGs in the KEGG pathway. TBtools was employed to construct heatmaps of the transcriptome [45].

### 2.6. Real-Time Quantitative PCR (RT-qPCR)

To verify the reliability of the transcriptomic RNA-seq data, we randomly selected some genes for RT-qPCR validation at different developmental stages. For the gene expression analysis, first-strand cDNA was synthesized from 1 µg of total RNA for each sample using the *TransScript*^®^ II All-in-One First-Strand cDNA Synthesis SuperMix for qPCR Kit (TRAN, Beijing, China). qRT-PCR was performed in a 96-well plate on a CFX96 Touch Real-Time PCR system (Bio-RAD, Hercules, CA, USA) using the *TransStart*^®^ Green qPCR SuperMix (TRAN, Beijing, China). The thermal conditions were 95 °C for 30 s, followed by 39 cycles of 95 °C for 15 s, 60 °C for 30 s, and 72 °C for 15 s. Then, gene-specific primers were designed based on multiple sequence alignment. The relative expression levels of the selected DEGs were calculated using the 2^−∆∆CT^ method [46]; specific information about the primers used in this study is shown in Appendix A. *Actin 7* [47] was used as an internal reference control.

### 2.7. Statistical Analysis

The data were analyzed and graphed using SPSS version 18.0. An independent-sample two-tailed Student’s *t*-test was used to analyze the significant of differences between the wild type and the *ms* mutant.

## 3. Results

### 3.1. Phenotype Investigation of MS Mutant

In this study, we report the discovery of a natural mutant in the WT rapeseed background (ZY511, Zhongyou511) that exhibits a remarkable phenotype. A homozygote of the mutant was obtained by self-crossing for six generations. Unlike the WT, which develops a single main stem, the mutant produces more than two branches, typically six to seven (Figure 1A,B). Importantly, the branches in the *ms* mutant were formed from the base of the plant. Additionally, the branching architecture of the *ms* mutant is characterized by small branch angles and a compact structure (Figure 1C,D). Given its distinctive phenotype, this mutant was named *ms* (multi-main-stem) rapeseed. To ensure the stable inheritance of this trait, we self-crossed the mutant for several generations to obtain homozygotes.

### 3.2. Abnormal Development of the SAM in MS Mutant

At 20 days after germination (DAG), two axillary buds had emerged in the WT (Figure 2A,B), whereas the *ms* mutant had multiple axillary buds at the base of the plant (Figure 2A,C). Thus, the number of main stems in the *ms* mutant significantly exceeded that of the WT, starting at 20 DAG. Shoot branching arises from the AMs in the axils of leaves, which are initiated by the SAMs [4]. Given that the *ms* mutant exhibited an increase in main stems, it was possible that SAM initiation was affected. To test this hypothesis, paraffin sectioning and scanning electron microscopy analysis of SAMs from the WT and *ms* mutant at 20 DAG were performed. The SAMs of WT exhibited a regular protuberance (Figure 2D,F). In contrast, the SAM in *ms* mutant exhibited an irregular shape and more than one SAM (Figure 2E,G). These results indicate a higher activity of the SAM in the *ms* mutant, which leads to the generation of multiple main stems.

### 3.3. Changes in Content of CKs in the SAM of the ms Mutant

CKs positively regulate the development of SAMs [9]. Specifically, we investigated the CK levels in the SAMs of both WT and *ms* mutant plants at 20 DAG. HPLC-MS analysis revealed that the levels of CKs were altered in the *ms* mutant compared to the WT. The most significant difference was observed in the levels of cis-zeatin riboside (czR) and trans-zeatin riboside (tzR) (Figure 2H). Among the active forms of CK, only isopentenyladenine (iP) was slightly increased in the *ms* mutant, whereas trans-zeatin (tZ) and cis-zeatin (cZ) were decreased. Compared to in the WT, the levels of the iP, CK ribosides czR and tzR were increased in the *ms* mutant by 21%, 47%, and 38%, respectively (Figure 2H). However, there was no significant change in the level of the nucleoside form of iP (iPR) in the *ms* mutant.

### 3.4. RNA-seq Analysis of the SAM in MS Mutant

To further elucidate the mechanism of the development of SAMs in the *ms* mutant, transcriptomic analysis was performed. We selected the SAMs of the WT and mutant at five developmental stages (15, 20, 25, 30, and 35 DAG) for transcriptome sequencing. First, we analyzed the reliability of the RNA-seq data. By using the FPKM values of the average of three replicates of each sample, we plotted a heatmap of the correlation between samples (Figure 3A), which showed that the coefficient of the correlation between each sample was above 0.8, suggesting high data consistency. Additionally, principal component analysis (PCA) of the samples showed that the replicates exhibited good consistency among different samples, meeting the requirements for further data analysis (Figure 3B).

### 3.5. Differential Transcriptional in Different Developmental Stages in the SAM of MS Mutant

The selection criteria for differentially expressed genes (DEGs) were set as follows: *p* value < 0.05, and |log_2_(fold change) | ≥ 1. The transcriptome of the SAMs of the *ms* mutant was compared with that of the WT at five developmental stages, and DEGs were identified. At 15 DAG, a total of 2685 DEGs were identified, including 1553 genes that were downregulated and 1132 genes that were upregulated. At 20 DAG, a total of 4782 DEGs were identified, of which 3420 were downregulated and 1362 were upregulated. At 25 DAG, a total of 5224 DEGs were identified, with 3058 downregulated genes and 2166 upregulated genes. At 30 DAG, a total of 3590 DEGs were identified, with 2261 downregulated genes and 1329 upregulated genes. At 35 DAG, 5460 DEGs were identified, of which 2269 were downregulated and 3193 were upregulated (Figure 3C, Appendix A). In addition, Venn diagram analysis of DEGs across these five developmental stages revealed that a total of 796 core genes were commonly regulated in both the WT and *ms* mutant (Figure 3D).

### 3.6. GO and KEGG Analysis of All DEGs

To further elucidate the functions of DEGs between the *ms* mutant and WT, Gene Ontology (GO) enrichment analysis and Kyoto Encyclopedia of Genes and Genomes (KEGG) pathway analysis were performed. For DEGs identified at 15 DAG, we observed 50 significantly enriched GO terms, with most DEGs being enriched in the molecular function “binding”, the cellular components “cell wall” and “apoplast”, and the biological process “translation” (Appendix A, Appendix A). Similarly, 48, 50, 50, and 49 significant GO terms were identified for DEGs at 20, 25, 30, and 35 DAG, respectively. Most DEGs were involved in the molecular function “DNA-binding transcription factor activity” (20 and 35 DAG), the biological process “response to chitin” (20, 30, and 35 DAG), “response to water deprivation” (25 DAG), and the cellular component “ubiquitin ligase complex” (20 DAG) and “apoplast” (25, 30, and 35 DAG) (Appendix A; Appendix A).

Furthermore, we identified the top 10 KEGG pathways for DEGs at 15, 20, 25, 30, and 35 DAG. The metabolism of amino acids, such as “β-alanine metabolism”, “valine, leucine, and isoleucine degradation”, and “alanine, aspartate and glutamate metabolism”, was significantly enriched, suggesting the potential involvement of these amino acids in the development of the *ms* mutant’s SAM. Notably, “plant hormone signal transduction” was enriched from the DEGs at 20, 30, and 35 DAG, indicating that the DEGs involved in this pathway also participate in the formation of multiple main stems in the *ms* mutant (Figure 4, Appendix A).

### 3.7. The Expression of CK-Related Genes Was Affected in the SAM of MS Mutant

Given the crucial role of CKs in regulating the development of SAMs, we focused on the transcriptional levels of genes involved in CK synthesis and metabolism pathways. The LONELY GUY (LOG) are CK-activating enzymes involved in CK synthesis, and the degradation of CK is performed by cytokinin oxidase/dehydrogenase (*CKX*) (Figure 5A). In this study, we detected 9 *LOG* genes (LOC106396898 (*LOG1*), LOC106352601 (*LOG1*), LOC106446715 (*LOG1*), LOC106356714 (*LOG4*), LOC106375311 (*LOG5*), LOC106453249 (*LOG5*), LOC106384114 (*LOG7*), LOC106440364 (*LOG8*), and LOC106364020 (*LOG8*)) and *CKX* (LOC106415860 (*CKX6*), LOC106406332 (*CKX6*), LOC106384861 (*CKX7*), LOC106406751 (*CKX7*), and LOC10636555 (*CKX7*)) with different expression in the *ms* mutant at different DAGs.

During the 15 DAG, before the formation of multiple stems in the *ms* mutant, there was no significant change in the expression of the *LOG* and *CKX* genes in the *ms* mutant compared to the WT. However, during the 20 DAG when the multiple stems of the *ms* mutant started to form, we found that the expression of *LOG8* was upregulated in the mutant but not in the WT. Conversely, the transcript levels of *CKX6* and *CKX7* were downregulated in the *ms* mutant but upregulated in the WT. The same trends were observed for these genes at 25 DAG. However, at 30 and 35 DAG, the expression of the *LOG* and *CKX* genes showed different patterns (Figure 5B, Appendix A). Moreover, we observed a significant upregulation of *IPT9*, a key gene involved in cytokinin (CK) biosynthesis [49], at 20 DAG (Appendix A). In terms of CK signal transduction, several members of the type-A ARABIDOPSIS REGULATOR (ARR) family, including *ARR3*, *ARR4*, *ARR6*, *ARR8*, *ARR16*, and *ARR17*, were also found to be significantly upregulated in the *ms* mutant at 20 DAG (Appendix A, Appendix A). These findings, combined with our previous investigations with regard to paraffin sections, CK content, and KEGG enrichment analysis, provide compelling evidence that the initiation of multi-main-stem formation in the *ms* mutant occurs primarily at 20 DAG.

Therefore, following 20 DAG, we selected *LOG7* (LOC106384114), *LOG8* (LOC106440364), *CKX6* (LOC106415860), *CKX7-1* (LOC106384861), *CKX7-2* (LOC106406751), and *CKX7-3* (LOC10636555) for RT-qPCR validation and further analysis. Both the transcriptome and RT-qPCR results showed that the expression of *LOG8* was upregulated in the *ms* mutant compared to the WT, whereas there was a downregulation of the expression of *LOG7* (Figure 5C). In contrast, the expression of *CKX6*, *CKX7-1*, *CKX7-2*, and *CKX7-3* was downregulated in the *ms* mutant relative to the WT (Figure 5C). Enzymes belonging to the *LOG* family play a crucial role in converting iPRMP (isopentenyladenine riboside 5’-monophosphate) into its active CK form, iP [35]. The catalysis of iP is carried out by CKXs [10]. Hence, the substantial downregulation of *CKX* genes and the concurrent upregulation of *LOG8* in the *ms* mutant have the potential to trigger an elevation in the levels of active cytokinin (iP) content.

## 4. Discussion

Multiple main stem formation is a crucial determinant of stem architecture in rapeseed and plays an essential role in improving vegetable yield. Despite its importance, the mechanism underlying the formation of multiple main stems in rapeseed remains largely unknown. In this study, we identified the *ms* mutant that exhibits 6–7 main stems (Figure 1). We hypothesized that the increased main branching was caused by abnormal development of the SAM. Further investigation revealed that the *ms* mutant had a greater number of SAMs with an irregular shape and increased axillary buds at the base than the WT at 20 DAG (Figure 2A–G). These results provide evidence that the increased number of main stems in the *ms* mutant is a result of an increased number of SAMs.

As the development of SAMs in the *ms* mutant was distinct from that of the WT by 20 DAG, we chose to perform transcriptome analysis at different time points, including the early germination stage (15 DAG), the day when SAM development started to produce abnormalities (20 DAG), and later time points (25, 30, and 35 DAG), to gain a comprehensive understanding of the SAM development in the *ms* mutant compared to the WT. At 15 DAG, there were a total of 2685 DEGs, much lower than the number observed at 20 DAG (4782), 25 DAG (5224), 30 DAG (3590), and 35 DAG (5460) (Figure 3C, Appendix A). Therefore, it is reasonable to conclude that, at least by 20 DAG, the development and gene expression of SAM in the *ms* mutant begin to undergo significant changes compared to the WT, which ultimately lead to the formation of multiple main stems at 60 DAG and 120 DAG (Figure 1).

Furthermore, we found that the “linoleic acid metabolism” and “α-linoleic acid metabolism” pathways were enriched in the *ms* mutant at 20, 25, 30, and 35 DAG (Figure 4). The jasmonic acid (JA) synthesis and signaling pathways are widely recognized for their pivotal role in plant defense against herbivory [50,51]. Therefore, the perturbation of these pathways in the *ms* mutant suggests a potential impact on its resistance to insects. Additionally, we observed that the “plant hormone signal transduction” pathway was enriched at least three time points (20, 30, and 35 DAG), suggesting that phytohormones also play a crucial role in the development of SAMs in the *ms* mutant (Figure 4).

Zhu et al. (2019) made a significant discovery regarding the role of CKs as critical regulators of SAM development in the *dt* (dou tou) mutant of *Brassica napus* L., which displayed an increased number of main stems. They found that the SAM of the *dt* mutant exhibited elevated levels of active CKs, including tZ and iP, as well as CK ribosides such as tzR and iPR. Furthermore, the levels of DHZR (dihydrozeatin riboside) and IP7G (N6-(Δ2-isopentenyl) adenosine-7-β-D-glucoside) were also increased in the SAMs of the *dt* mutant [52]. However, in our study, we observed modestly elevated levels of active CK (iP) and CK ribosides (czR and tzR) in the SAMs of the *ms* mutant compared to the WT (Figure 2H), albeit with differences from the *dt* mutant.

Through transcriptome and RT-qPCR analysis, we have revealed intriguing insights into the molecular mechanisms underlying the differences shown by the *ms* mutant. Notably, we observed an elevation in the expression of the CK synthesis gene *LOG8*, whereas the CK degradation gene *CKX6* and three *CKX7s* were downregulated in the ms mutant compared to the WT (Figure 5B,C). In addition to *LOG* and *CKX*, we also investigated the transcript levels of *IPTs* and genes involved in CK signal transduction in the *ms* mutant. Interestingly, we found minimal significant changes in gene expression at 15 DAG between the *ms* mutant and WT. Furthermore, the expression levels of CK receptor *AHKs* [53], *AHPs* [54], and *type-B ARRs* [55] exhibited relatively limited alterations in the *ms* mutant across all DAG (Appendix A). However, at 20 DAG, a considerable number of genes, particularly type-A ARRs, were significantly upregulated in the *ms* mutant (Appendix A). This observation aligns with our hypothesis that the alteration in the SAM of the *ms* mutant commenced at 20 DAG. Surprisingly, at 35 DAG, many type-A ARRs exhibited upregulation, whereas some showed slight downregulation at 30 DAG (Appendix A). These findings appear contrary to the expected role of type-A ARR proteins as negative regulators of CK signaling and SAM function. Nevertheless, previous research by Müller et al. (2015) demonstrated that the hextuple type-A *arr3,4,5,6,7,15* mutant of Arabidopsis displayed reduced rosette branching and bud activation compared to the WT [56]. Therefore, further investigation is warranted to elucidate the role of CK in SAM development and to determine whether type-A ARRs are involved in regulating the formation of multi-main stems in the *ms* mutant.

In addition to CK, shoot branching is also regulated by auxin and strigolactones (SLs) [57,58,59,60,61]. Previous studies have shown that CK, auxin, and SL signals can interact with each other [7,62,63]. More recently, Tang et al. (2020) identified a new high-density pod mutant of *Brassica napus* that displayed multiple stems and higher levels of indole-3-acetic acid (IAA) in the SAM [64]. In this study, we found that at 20 DAG, the GO enrichment analysis revealed that 45 upregulated and 12 downregulated DEGs in the *ms* mutant were involved in the “auxin-activated signaling pathway”, and “response to auxin” was among the pathways enriched from the 2 upregulated and 43 down-regulated DEGs in the *ms* mutant (Appendix A, Appendix A), indicating that the auxin pathway might be involved in the formation of multiple main stems in the *ms* mutant.

In our investigation, we focused on CK signaling and observed a slight increase in iP content in the SAM of *ms* mutant, but no significant changes in other active CK types. This observation does not strongly support the hypothesis that CK regulates SAM development and shoot branching in the *ms* mutant. Hence, it is imperative to delve into the alterations in other phytohormones within the *ms* mutant in future investigations. Furthermore, determining whether auxin or other phytohormones play a role in regulating branching in the *ms* mutant through CK signaling necessitates further exploration.

Besides phytohormones, sugar also promotes cell division, and it plays a role in plant branching. Salam et al. (2017) reported that the branching of potato (*Solanum tuberosum*) was induced by the application of an exogenous sugar (sucrose, fructose, or glucose) [65]. Our KEGG analysis revealed that, at 20 and 25 DAG, when the SAM was abnormal in the *ms* mutant, the DEGs between the *ms* and WT were enriched in the “amino sugar and nucleotide sugar metabolism” and “starch and sucrose metabolism” pathways (Figure 4B,C, Appendix A), suggesting that the sugar signaling may be involved in the formation of multiple main stems in the *ms* mutant. Thus, shoot branching is a complex quantitative trait that is under polygenic control and is regulated by phytohormones and external environmental factors; much research on CK, SLs, auxin, and sugar is needed to fully understand the increased branching in the *ms* mutant.

The identification of a mutation site in a newly discovered natural mutant typically requires the use of various tools and approaches. One of the most powerful tools in identifying loci or key genes associated with agronomic traits is the genome-wide association study (GWAS) [66,67]. In rapeseed, as with many other crop plants, most agronomic traits are quantitatively inherited and controlled by quantitative trait loci (QTLs) [68,69,70,71]. QTLs related to branch number have been identified in rapeseed [72,73,74,75]. Based on GWAS, numerous loci related to branch number have been identified on almost all rapeseed’s chromosomes, including A01, A03, A07, C04, C07, and C09 [76,77,78,79,80]. In addition to GWAS, Li et al. (2020) identified a major QTL related to branching (*shoot branching 9*, *qSB.A09*) on the chromosome A09 in *Brassica rapa* L. ssp. *Chinensis* by integrating QTL mapping with BSA-seq (bulked segregant analysis) [81]. Given that 43 QTLs were identified for MMS in rapeseed, and six candidate genes related to the formation of MMS were obtained from QTL mapping and using the gene-fishing technique [40], it will be worthwhile to explore the major QTLs or candidate genes associated with multiple main stems in the *ms* mutant using these methods in the future.

In the context of new rapeseed mutants, the high-density pod mutant (*dpt247*) not only exhibits multiple stems but also shows a reduced plant height and primary branch length, along with a significantly increased number of pods on the main inflorescence [64]. Similarly, the dou tou (*dt*) mutant displays multiple flowers from a single peduncle, and the maturation of dt siliques is considerably slower compared to that of the WT, in addition to multiple main stems being present [52]. These findings suggest that gene mutations in new mutants can lead to diverse changes beyond a single phenotype. Therefore, conducting detailed observations and analyses of the *ms* mutants in the future will be essential for a comprehensive understanding of their characteristics.

## 5. Conclusions

Our study sheds light on the mechanism underlying the formation of multiple main stems in rapeseed. Specifically, we found that the abnormal development of the SAM in the *ms* mutant is closely linked to the increased number of main stems. Through comprehensive transcriptome analysis conducted on both WT and *ms* mutant at various germination stages, we made a significant discovery that the aberrant development of the *ms* mutant’s SAM initiates at 20 DAG. Furthermore, our investigation unveiled noteworthy alterations in the expression profiles of genes associated with the CK signaling pathway within the SAM of the *ms* mutant, when compared to the WT. Intriguingly, despite conducting HPLC-MS analyses, we did not detect any substantial changes in the CK levels present in the *ms* mutants. However, only a subtle upregulation of iP, tzR, and czR was observed. This study not only provides insights for breeding new rapeseed varieties with multiple main stems but also provides valuable resources for future research on SAM development and shoot branching in rapeseed. However, the loci responsible for the multi-main-stem phenotype in the *ms* mutant remain unknown, and further GWAS and genetic fine mapping will be necessary to identify the major QTLs or candidate genes associated with multiple main stems in the *ms* mutant.

## Figures and Tables

**Figure 1 genes-14-01396-f001:**
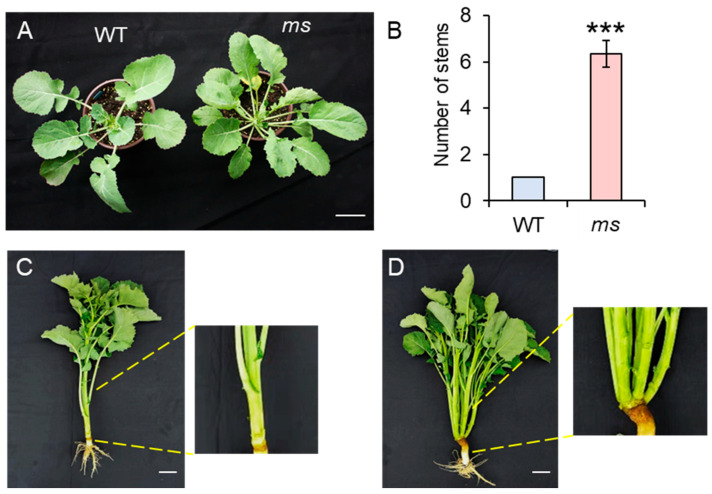
(**A**) The phenotypes of wild-type (WT) and multi-stem (*ms*) mutants after germination at 60 days. Bar = 5 cm. (**B**) The numbers of the main stems of the WT and *ms* mutant. Values are the means (±SDs) (two-tailed Student’s *t*-tests; *** *p* < 0.001; *n* = 10). (**C**) The phenotypes of the WT and (**D**) *ms* mutant after germination at 120 days. Bar = 10 cm. The yellow lines indicate the enlargement of the base of the stem in the WT and *ms* mutant plants.

**Figure 2 genes-14-01396-f002:**
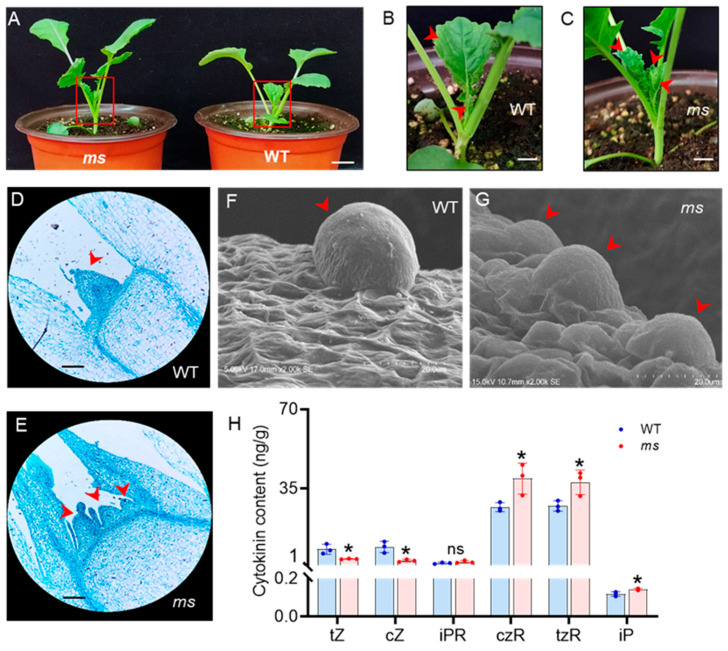
The development of the shoot apical meristem in the WT and *ms* mutant. (**A**) Phenotypes of the wild-type (WT) and multi-stem (*ms*) mutant plants after germination at 20 days; bar = 2 cm. (**B**) Close-up views of the axillary buds (red arrowheads) in the WT and (**C**) *ms* mutant are shown in the red panels in (**A**), respectively; bar = 0.5 cm. (**D**) Paraffin section of the shoot apical meristems (SAMs, red arrowheads) of the WT and (**E**) *ms* mutants after germination at 20 days; ×100 magnification; bar = 100 μm; *n* = 10. (**F**) Scanned images of the SAMs (red arrowheads) in the WT (voltage: 5 kV; magnification: ×2000; and cross-section: 17 mm) and (**G**) *ms* mutant after germination at 20 days (voltage: 15 kV; magnification: ×2000; and cross-section: 10.7 mm); bar = 20 µm; *n* = 10. (**H**) The cytokinin contents in the SAMs of the WT and *ms* mutants after germination at 20 days. tZ: trans-zeatin; cZ: cis-zeatin; iPR: isopentenyladenine riboside; czR: cis-zeatin riboside; tzR: trans-zeatin riboside; iP: isopentenyladenine. Values are the means (±SDs). Significant differences between the WT and *ms* mutant are indicated (two-tailed Student’s *t*-tests; * *p* < 0.05; NS, not significant; *n* = 3).

**Figure 3 genes-14-01396-f003:**
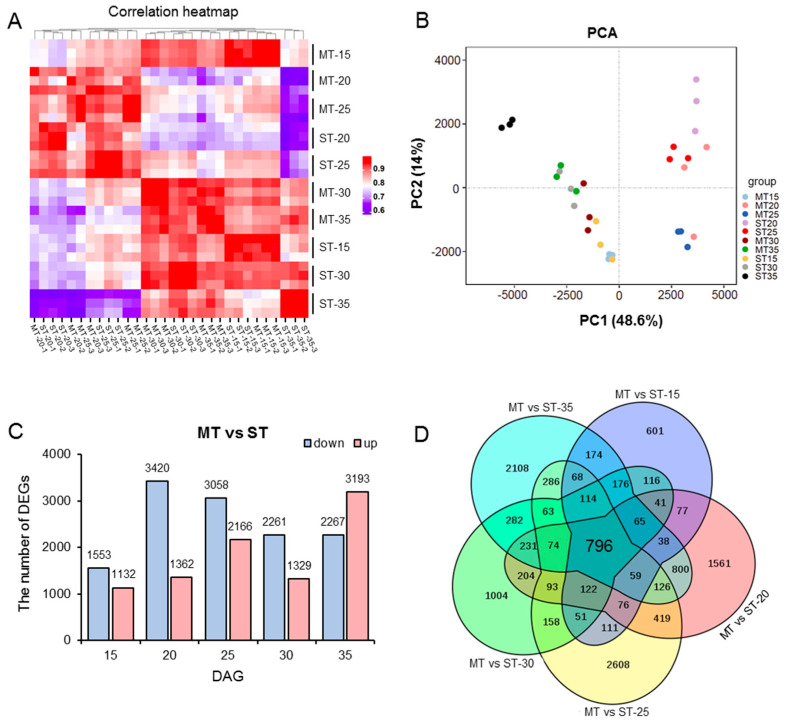
Global transcript analysis of WT compared with *ms* mutant. The SAMs of the WT and *ms* mutant were collected at 15, 20, 25, 30, and 35 days after germination (DAG). (**A**) Heatmap of Pearson correlation between samples according to gene expression values. (**B**) Principal component analysis (PCA) of the 10 groups of transcriptome data. (**C**) The number of differentially expressed genes (DEGs) between the *ms* mutant (M) and WT (S) at different DAG. (**D**) Venn diagrams indicating the numbers of common and specific DEGs identified between the *ms* mutant (M) and WT (S) at different DAG. ST: time after germination of single-stem plants (WT); MT: time after germination of multi-stem (*ms*) mutants. Complete data can be found in Appendix A.

**Figure 4 genes-14-01396-f004:**
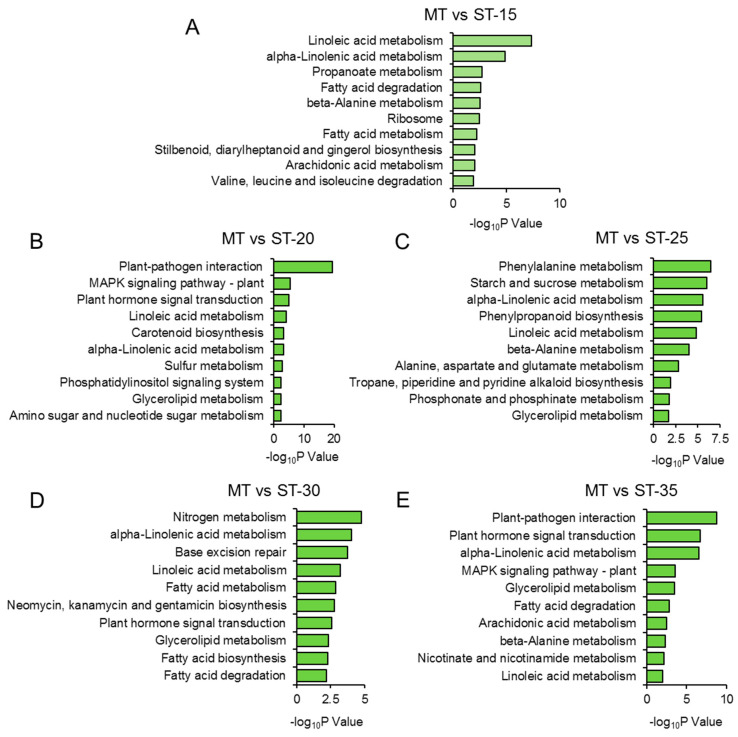
KEGG enrichment analysis of DEGs between *ms* mutant and WT. Top 10 KEGG pathways of DEGs in *ms* mutant compared with WT at (**A**) 15, (**B**) 20, (**C**) 25, (**D**) 30, and (**E**) 35 days after germination. ST: time after germination of single-stem plants (WT); MT: time after germination of multi-stem (*ms*) mutants. Complete data can be found in Appendix A.

**Figure 5 genes-14-01396-f005:**
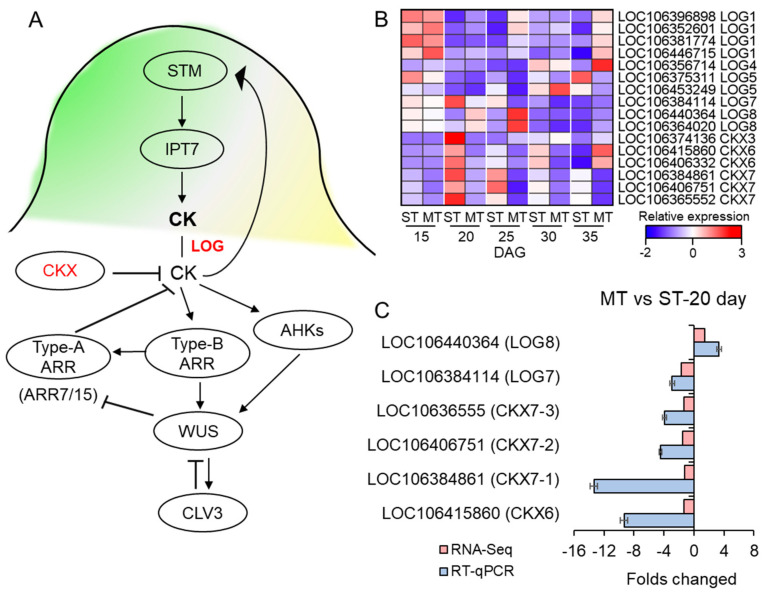
The expression of genes involved in cytokinin signaling. (**A**) The gene networks involved in the cytokinin (CK) signaling pathway. *LOG* positively regulates CK synthesis, and *CKX* negatively regulates the synthesis of CK. CK signaling positively regulates *WUS* in a manner that is dependent on type-B ARRs; type-B ARRs directly activate type-A ARRs, which act as negative-feedback regulators of CK signaling, and WUS represses type-A ARRs (*ARR7/15*). CK also positively regulates *WUS* transcription via cytokinin receptors called *AHKs*. Adapted from [48]. (**B**) Heatmap of the *LOG* and *CKX* gene family in WT (S) and *ms* mutant (M) at 15, 20, 25, 30, and 35 days after germination (DAG). The colors correspond to the average FPKMs of three replicates, ranging from blue (low expression) to red (high expression). Complete data can be found in Appendix A. (**C**) RT-qPCR and the RNA-seq indicated the relative expression (fold change) of the *LOG* and *CKX* genes in the *ms* mutant (M) relative to the WT (S) at 20 days after germination. ST: time after germination of single-stem plants (WT), MT: time after germination of multi-main-stem (*ms*) mutants.

## Data Availability

All data were included in the manuscript.

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
