# Peer review of "Transcriptomic Profiling of Shoot Apical Meristem Aberrations in the Multi-Main-Stem Mutant (ms) of Brassica napus L."

_genes, 2023, doi:10.3390/genes14071396_

Round 1

Reviewer 1 Report

Comments

The ms Genes-2377032 by Wang et al. “Cytokinin signaling are required for multi-main stems development in Brassica napus L.” describes a native rapeseed mutant forming multiple stems unlike WT plants having only one stem. For this study, the mutants were subjected to self-crossing for six generations, obtaining the homozygous status. Three types of experiments were conducted: meristem visualization using also SEM; cytokinin (CK) quantification and gene expression profiling. The authors concluded that their study revealed difference in meristem organization between WT and mutant plants which they ascribe to increased CK content in mutant meristems. High CK level in turn is supposed to affect gene expression patterns. Marked differences of the latter were indeed demonstrated in respective genome-wide experiments.

However, despite the theme is really interesting, this study leaves strong doubts concerning conclusions related to cytokinin signaling and gene expression. In the Introduction, the authors characterize CKs as “cytokinins (CKs), a group of N6 adenines that include isopentenyladenine (IP), trans-zeatin (tZ), cis-zeatin (cZ), and dihydrozeatin (DHZ) …  “ (lines 46-48). This definition is partly adequate, partly not. What is correct, these are terms of 4 nucleobases (tZ, cZ, iP and DZ) which are only active CKs in vivo. This eventually became clear rather recently (in 2015), so the respective reference should be added. But CKs are not a group of N6 adenines (they are N6-derivatives of adenine) and they unify not only 4 plant CKs but also many tens of artificial CKs of different structures. This is general definition, but the reviewing study deals with natural CKs only. The data on their content in SAM together with the content of corresponding CK-ribosides are demonstrated in Figure 2H. The graphic shows that the content of active CKs involved in signaling was not increased in mutants versus WT plants, but, on the contrary, was seemingly reduced. What was increased, these were contents of some CK-ribosides, but ribosides do not take part in signaling and were rightly not mentioned in the Introduction. So, the main conclusion of the ms turns out to be disproved by direct authors’ data, thus the general title, paragraph title (line 177) and some other similar sentences are contradicting to actual experimental data.

Meanwhile, despite the CK contents in SAM of WT and mutant plants were evidently very similar, more intense CK signaling in mutants can nevertheless occur due to presumable higher activity of CK receptors. These receptors in B. napus were studied in detail yet in 2015 (Kuderova et al., J.Exp.Bot.). But this ms does not consider rapeseed CK receptors or encoding genes, although authors should have genome-wide data about gene expression in SAM of WT and mutant lines. These gene expression databases can provide also an independent indicator of CK signaling level. I mean here the ARR-type A primary response genes whose expression is greatly upregulated by CK signaling. But here too, no any attempt to compare and analyze the expression of such genes in WT and mutant plants was described. Thus, this important part of the work is clearly far from being completed.

The 3rd part of experimental work also seems to lack necessary scientific background. When one compares the expression profiling between WT and mutant lines, one should be sure that apart mutation, all other plant traits are virtually identical, at least both lines belong in our case to the same rapeseed variety. There exist to date a lot of rapeseed cultivars, more information about them can be found on several websites (www.rapool.de3, www.roth-agrar.de4  and www.dsv-saaten.de/raps/winterraps/sorten). But in the ms, I was not able to find any indication on the variety of any of plant group used, that probably signify that the varieties may be different and/or simply unknown.

If this is really the case (it is quite probable), the DEG analysis loses its big sense because cultivar-related gene expression specificity is assumed to largely dominate over the mutation specificity. Homozygous state of mutant genotype vs heterozygous state of WT control can also be the cause of pronounced DEG which has nothing of common with mutation-related DEG per se.

The English mistakes do not improve the overall impression of this ms. The obvious grammar mistake localizes even in the title of the ms. 

Reviewer 2 Report

In the present manuscript the authors describe the characterization of a new mutant with multiple main stems named multiple stem (ms) through the use of histological techniques and scanning electron microscopy. Subsequently, they performed the quantification of cytokinins in the SAM as well as a transcriptomic analysis, finding that the ms mutant has a higher content of cytokinins in the SAM and this correlates with the expression levels of genes involved in cytokinins metabolism (biosynthesis and degradation). In general, this manuscript presents good information that support the characterization of the ms mutant. However, there are certain aspects that need to be improved.

1. The title of the manuscript refers to the participation of cytokinin signaling in the multiple stem characteristic of the ms mutant. However, during the work only genes related to cytokinin metabolism were analyzed and no data is shown on cytokinin signaling genes, so the title should be modified.

2. Information is missing from the introduction about what is already known at least in Brassicaceae about the role of cytokinins in the SAM (refer to Figure 5A).

3. The materials and methods section is poorly described, so it needs to be improved:

a. The genetic background from which the ms mutant was isolated must be indicated.

b. The culture medium and the characteristics of the medium in which the seeds were germinated must be indicated.

c. The technique or instrument used to tissue sections should be indicated, as well as the dyes used to stain the SAM sections. In the cited bibliography different sections techniques and different stains are mentioned, please specify which ones were used in this work.

d. The treatment given to the samples that were analyzed with SEM must be indicated (critical point, precious metal coating or fresh tissue analysis).

e. In reference (30) different analytical methods are reviewed, please specify the analytical method used during this work.

f. It should be indicated How was cDNA synthesis performed? What kit was used?

g. The fluorometric method used in the real-time PCR must be indicated (SYBER GREEN or other).

h. It is not clear if the actin7 primers were designed by the authors or have been previously reported.

i. Since the primers were generated by the authors, it would be necessary to know the efficiency of the primers.

j. Specify which statistical methods or tests were used during this work.

4. Figures need to be improved

FIGURE 1B. The data in the graph must be statistically analyzed to determine the statistical difference between the WT and the ms mutant. Furthermore, such statistical difference should be noted in the figure.

FIGURE 2D and 2E. Add the size reference bar. In addition, the SAMs must be indicated with arrows.

FIGURE 2F and 2G. Add the size reference bar, since only some points are observed. In addition, the SAMs must be indicated with arrows.

FIGURE 5A. Missing information in this figure, you must add to the ARR TYPE B, the repression of ARR7/15 through WUS, the repression of CK through the ARR TYPE A. Furthermore, the LOG genes should be below IPT7.

GENERAL

1. Line 79: “named Multiple Stem (MS) rapeseed”, mutant names are written in lowercase italics “multiple stem (ms)”. Also, in some cases “multi-stem” is written instead of “multiple-stem” as in line 188 (Caption of figure 2A). Correct all over the text.

2. Line 95: “DAG20-old”, the number is written before, 20 days after germination (20 DAG) , correct in all the text.

3. Lines 166-167: “At the 20 days after germination (DAG), there are two axillary buds emerged in WT, while the MS mutant has multiple axillary buds at the basal of the plant (Figure 2A and 2B)”. Indicate with arrows the axillary buds in Figures 2A and 2B.

4. Lines 173-174: “The SAM of WT exhibited a regular bulge (Figure 2D and 2F). In contrast, the SAM in MS mutant exhibited an irregular shape and more than one SAM (Figure 2E and 2G)”. Indicate with arrows the different SAMs in figures 2D, 2E, 2F and 2G.

5. Line 239: “DAN-binding transcription factor activity”, change to “DNA-binding transcription factor activity”.

6. Line 257: “significantly higher CKs level in the SAM of MS mutant compared to WT (Figure 2C)”, it refer to figure 2H instead of 2C.

7. Lines 257-258: “we focused on the transcriptional levels of genes involved in CK synthesis and metabolism pathways.” What is the expression profile of the other genes involved in CK biosynthesis, such as the ISOPENTENYLTRANSFERASEs (IPTs) genes?

8. Line 260: “cytokinin oxidase/dehydrogenase”, gene names are written in capital letters and italics.

9. Lines 261-262: “The CK signaling pathway is activated by LONELY GUY (LOG), which releases free bases from nucleotide forms.” LOG genes do not activate CK signaling, which is activated by the hormone that is released by LOG genes.

10. Lines 263-268: “In this study, we detected 14 LOG (LOC106396898 (LOG1), LOC106352601 (LOG1), LOC106446715 (LOG1), LOC106356714 (LOG4), LOC106375311 (LOG5), LOC106453249 (LOG5), LOC106384114 (LOG7), LOC106440364 (LOG8), LOC106364020 (LOG8)) and CKX (LOC106415860 (CKX6), LOC106406332 (CKX6), LOC106384861 (CKX7), LOC106406751 (CKX7), LOC10636555 (CKX7)) genes with different expression in the MS mutant at different DAGs.”

In this study, we detected 9 LOG genes (LOC106396898 (LOG1), LOC106352601 (LOG1), LOC106446715 (LOG1), LOC106356714 (LOG4), LOC106375311 (LOG5), LOC106453249 (LOG5), LOC106384114 (LOG7), LOC106440364 (LOG8), LOC106364020 (LOG8)) and 5 CKX genes (LOC106415860 (CKX6), LOC106406332 (CKX6), LOC106384861 (CKX7), LOC106406751 (CKX7), LOC10636555 (CKX7)) with different expression level in the MS mutant at different DAGs.

11. In addition to the CK metabolism genes, what other CK-related genes were differentially expressed between the WT and the mutant? What is the expression level of genes involved in CK signaling such as AHPs, AHKs, ARR type A and B?

12- Lines 282-284: “Both the transcriptome and RT-qPCR results showed that the expressions of LOG8 were up-regulated in MS mutant compared to WT, while there was no significant difference in the expression of LOG7”. What statistical test was performed to determine that there was no significant difference in the LOG7 expression level?

13. Lines 322-323: “These pathways are associated with jasmonic acid (JA) synthesis, which may be related to the wounding caused by collecting the samples.” If the activation of these pathways were related to the damage caused during collection, they should not be enriched since both wt and mutant were collected in the same way.

14. The discussion section could be improved if the references ZHU et al., 2019 Transcriptome analysis of the irregular shape of shoot apical meristem in dt (dou tou) mutant of Brassica napus L. and TANG et al., 2020 A recessive high-density pod mutant resource of Brassica napus are added and the differences and similarities in these studies are discussed, regarding the CK metabolism genes that were analyzed, the type and concentration of Cks that were quantified in the SAM.

At least a moderate editing of English language is needed.

Reviewer 3 Report

The manuscript authored by Wang et al. discovered a naturally occurring mutant of rapeseed with a stable inheritance of a multi-stem trait, leading to increased main stem numbers. The mutant exhibited altered gene expression and higher levels of cytokinins, providing insights into the mechanism behind multiple main stem formation and information for breeding multi-main stem rapeseed vegetable varieties. In general, the study is well designed and implemented. The conclusions are logical and well-supported based on the results. I have the following minor comments which could be addressed to improve the readership.

Please provide more information about this natural mutant. What is the origin of this mutant? How do you know this mutant from a germplasm collection was naturally derived from the WT in this study?

In Figure 4, Were all those DEGs between the WT and MS mutant compared at the same stage (DAG)? In other words, the DEGs for KEGG analysis in Figure 4A are differentially expressed genes between WT and MS at DAG 15? What does the MT vs ST-15 mean? Please label each panel according to the description in the figure legend and label them clearly if they are stage-specific comparisons. How are the DEGs among those five stages (Days after germination) in the WT or MS mutant? Inter-stage comparison could potentially tell us more about the temporal transcriptomic dynamics of SAM development.

It is well written. Some minor grammar errors can be corrected.

Round 2

Reviewer 1 Report

The corrected version did not correct the main flaw of this manuscript, namely the lack of convincing arguments in favor of the participation of cytokinins in the manifestation of the mutant phenotype. On the contrary, the data presented in Fig. 2 argue in favor of the nonparticipation of cytokinins in the induction of multi-main stem formation. Contrary to the authors' claims, Fig. 2 indicates not an increase but a general decrease in the content of active cytokinins through a clear drop in tZ and cZ. A microscopic increase in iP content, if any, would have no effect on rapeseed physiology. Moreover, this minor increase is many times overridden by the decrease in the concentration of tZ, the most active cytokinin in rapeseed (see Kuderova et al. 2015). I am wondering that contrary to logic and their own results, the authors continue to insist on the leading role of cytokinins in this phenomenon and even put this thesis in the title. However, the relevance of this thesis can be tested experimentally, for example, by treating seedlings of common rapeseed with cytokinin and then estimating the number of stems after such treatment. Or if the authors claim to have found genes that are likely inducers of the multi stem phenotype, then experiments should first be performed with the knocking out of these genes and/or with their overexpression in plant genome. Only if such experiments yield positive results, the term "cytokinins" can be put in the abstract and the title of the article. In the meantime, the scientific significance of the work, despite all the beautiful pictures taken from textbooks, is essentially zero.

English is satisfactory

Reviewer 2 Report

Dear Authors

You have made substantial changes by responding to almost all the comments made in the first round of revision. However, there is a point that I consider to be important and which was ignored, this refers to point “i” in the materials and methods section.

i. Since the primers were generated by the authors, it would be necessary to know the efficiency of the primers.

When primers are reported for the first time, it is important to shown, at least to the reviewers. The information regarding the efficiency and the dissociation curve. Efficiency is an essential parameter to be able to calculate the relative expression level of genes using the 2−ΔΔCT the method. The dissociation curve is used to verify that a single fragment was amplified (Real-time PCR handbook; Livak and Schmittgen, 2001).

Also, there are some points that need to be clarified.

1.- Lines 114-115. “Thus, this study lays the foundation for understanding the role of CK in SAM development”. Currently, important information regarding the role of CKs in the SAM is already available.

2.- Lines 148-149. “the wax block was removed from the embedding frame and repaired”. What do they mean by repaired?

3. Lines 162-164. “The sections were then put into a plant solid green staining solution for 6~20 s, before undergoing anhydrous ethanol three-cylinder dehydration. The sections were placed into three cylinders of xylene for 5 min. ”. What is “plant solid green staining solution” and “three-cylinder dehydration”?

4. There are still some times when the name of the mutant is written without italics as in line 318.

English has been substantially improved
